# HyperIMTS: Hypergraph Neural Network for Irregular Multivariate Time Series Forecasting

**Boyuan Li** [1]    **Yicheng Luo** [1]    **Zhen Liu** [1]    **Junhao Zheng** [1]    **Jianming Lv** [1]    **Qianli Ma** [† 1]

## Abstract

Irregular multivariate time series (IMTS) are characterized by irregular time intervals within variables and unaligned observations across variables, posing challenges in learning temporal and variable dependencies. Many existing IMTS models either require padded samples to learn separately from temporal and variable dimensions, or represent original samples via bipartite graphs or sets. However, the former approaches often need to handle extra padding values affecting efficiency and disrupting original sampling patterns, while the latter ones have limitations in capturing dependencies among unaligned observations. To represent and learn both dependencies from original observations in a unified form, we propose HyperIMTS, a **Hyper**graph neural network for **I**rregular **M**ultivariate **T**ime **S**eries forecasting. Observed values are converted as nodes in the hypergraph, interconnected by temporal and variable hyperedges to enable message passing among all observations. Through irregularity-aware message passing, HyperIMTS captures variable dependencies in a time-adaptive way to achieve accurate forecasting. Experiments demonstrate HyperIMTS's competitive performance among state-of-the-art models in IMTS forecasting with low computational cost. Our code is available at https://github.com/qianlima-lab/PyOmniTS.

## 1. Introduction

Multivariate Time Series (MTS) are prevalent in various fields such as healthcare, weather, and biomechanics (Zhang et al., 2023b; Shukla & Marlin, 2021). While MTS forecast-

ing has been extensively studied (Nie et al., 2022; Zhang & Yan, 2022; Zhou et al., 2021; Yu et al., 2024; Yi et al., 2023), these methods typically assume the input to be fully observed and pay less attention to potential sensor malfunctions, varying sampling sources, or human factors in reality. These factors can result in Irregular Multivariate Time Series (IMTS), which are characterized by irregular time intervals within each variable and unaligned observations across variables. These characteristics make it challenging to provide accurate forecasting on IMTS for informed decision-making and planning.

In studies for IMTS, one category of methods pads the series in the sample space to have same length across variables (Rubanova et al., 2019; Zhang et al., 2021; Tashiro et al., 2021; Zhang et al., 2023a; 2024), as illustrated in Figure 1 (a) and (b). Padded series are either time-aligned or patch-aligned, allowing for effective capturing of dependencies along both temporal and variable dimensions. However, such padding scheme can increase the amount of data to be processed, primarily because observation timestamps for different variables can span widely separated time periods. Furthermore, some methods rely on relative positions of observations instead of timestamps to represent temporal information, which might disrupt the original sampling pattern. It should be noted that padding refers to adding zeros or predicted values in the sample space before inputting IMTS into the neural network, differing from the continuous latent space used in ODE-based methods (Rubanova et al., 2019; Biloš et al., 2021; Mercatali et al., 2024).

Instead of relying on padded series, another category of IMTS analysis methods proposes using sets or bipartite graphs to represent original IMTS samples, as illustrated in Figure 1 (c) and (d). Set-based methods view observations as unordered tuples in a set (Horn et al., 2020), while bipartite graph approaches represent channels and timepoints as disjoint nodes connected by edges (Yalavarthi et al., 2024), both of which only express observed values for high efficiency. However, sets typically do not account for correlations among observations, and bipartite graphs are unable to propagate messages between variables without shared timestamps. These methods still have limitations in capturing dependencies on original IMTS.

---

[1]School of Computer Science and Engineering, South China University of Technology, Guangzhou, China. Correspondence to: Qianli Ma <qianlima@scut.edu.cn>.

*Proceedings of the 42nd International Conference on Machine Learning*, Vancouver, Canada. PMLR 267, 2025. Copyright 2025 by the author(s).

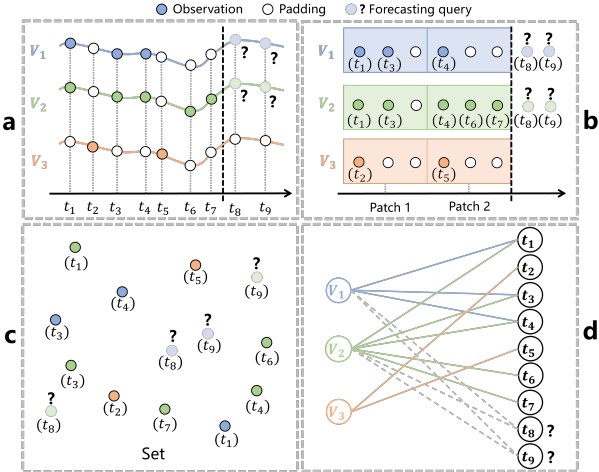

*Figure 1.* Existing methods for processing IMTS sample. (a) Canonical padding approach, which significantly increases the amount of data. (b) Patch-aligned padding approach, which also increase the amount of data. (c) Set views all observations as its unordered items. (d) Bipartite graph uses observation edges to connect variable and time nodes. It cannot model dependencies between variables without aligned observations, like $V_2$ and $V_3$, which require shared timestamps.

To represent original observations and comprehensively capture their dependencies, we propose HyperIMTS, a hypergraph neural network for IMTS forecasting. HyperIMTS represents observations and their dependencies in a unified hypergraph, transforming the IMTS forecasting task into the node prediction task. It views observations as nodes in the hypergraph, each associated with its corresponding timestamp and variable. Observations belonging to the same variable are connected by a multi-connected hyperedge, just as observations sharing the same timestamp. Temporal dependencies are captured through node-to-hyperedge and hyperedge-to-node message passing, while variable dependencies further benefit from hyperedge-to-hyperedge connections. Moreover, given the difficulty in capturing variable correlations under different time alignment situations, we calculate time-aware similarities using nodes and overall similarities using hyperedges to realize irregularity-aware learning of variable dependencies.

Our major contributions are summarized as follows:

- We propose a new hypergraph modeling approach to represent both observed values and their dependencies in IMTS, which does not require padding and remains extensible for dependency learning.

- Based on the hypergraph representation, we propose HyperIMTS, a hypergraph neural network for the IMTS forecasting task. It leverages timestamp informa-

tion preserved in the graph to adaptively capture both time-aware and overall variable dependencies, enabling irregularity-aware learning and accurate forecasting.

- We build a unified, extensible, and highly flexible code pipeline for fair IMTS forecasting benchmarking across time series models from various fields and tasks, covering twenty-seven state-of-the-art models and five IMTS datasets. Extensive empirical results demonstrate the low forecast error and high efficiency of HyperIMTS.

## 2. Related Work

### 2.1. Irregular Multivariate Time Series Modeling

Existing efforts on IMTS mainly focus on classification (Che et al., 2018; Shukla & Marlin, 2020; Zhang et al., 2021; 2023a; Horn et al., 2020; Shukla & Marlin, 2018) and imputation tasks (Che et al., 2018; Rubanova et al., 2019; Shukla & Marlin, 2020; Tashiro et al., 2021). In recent years, an increasing number of studies have paid attention to IMTS forecasting (Tashiro et al., 2021; Schirmer et al., 2022; Yalavarthi et al., 2024; Zhang et al., 2024; Mercatali et al., 2024). From a data preprocessing perspective, existing works on IMTS can be broadly categorized into padding and non-padding methods. The former ones typically represent input time series as matrices with temporal and variable dimensions, and they design model components to learn dependencies along both dimensions (Shukla & Marlin, 2020; Tashiro et al., 2021; Schirmer et al., 2022). Models based on RNNs (Che et al., 2018), ordinary differential equations (ODEs) (Rubanova et al., 2019; Biloš et al., 2021; Mercatali et al., 2024), transformers (Zhang et al., 2023a), and graph neural networks (Zhang et al., 2021; Luo et al., 2024; Zhang et al., 2024) are commonly used. While these models can achieve outstanding performance, they often require handling more input data, which can affect the efficiency during preprocessing and training. The other non-padding approaches use bipartite graph (You et al., 2020; Yalavarthi et al., 2024) or set (Horn et al., 2020) to represent IMTS. Their sparse representations can handle IMTS samples without padding, but their model architectures restrict the ability to capture dependencies in the original IMTS.

### 2.2. Using Graphs for MTS

Graphs are increasingly popular in time series analysis (Yi et al., 2023; Wen et al., 2023; Wu et al., 2020; Cini et al., 2021; Han et al., 2024), which usually represent variables as nodes and their dependencies as edges. Although this type of graph demonstrates great potential for learning multivariate dependencies, it requires time-aligned or patch-aligned samples for input encoding, which increases the amount of data for IMTS samples. The hypergraph is a variant where

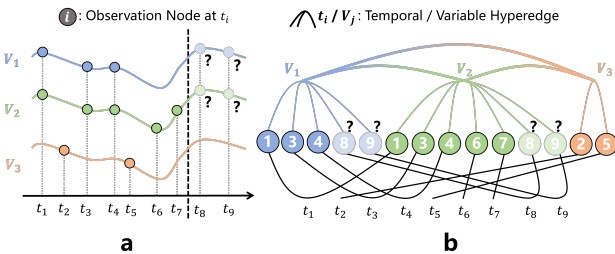

*Figure 2.* Illustration of the proposed efficient hypergraph representation for IMTS. (a) Original IMTS sample, where $V_1$, $V_2$, and $V_3$ represent three different variables. (b) The corresponding hypergraph representation. From top to bottom, variable hyperedges, observation nodes, and temporal hyperedges are displayed. Each observation node connects to the associated variable hyperedge above and the temporal hyperedge below. Gradient color lines between hyperedges indicate hyperedge-to-hyperedge message passing.

hyperedges can connect multiple nodes, facilitating more types of message passing. Its ability to represent data at different granularities and model their complex interactions has attracted researchers in various fields (Zhou et al., 2006; Yan et al., 2020; Gao et al., 2023; Yu et al., 2012). Hypergraph neural networks are still in their early stages for time series analysis (Shang et al., 2024; Luo et al., 2022; Li et al., 2022; Sawhney et al., 2021). They usually assume inputs to be fully observed, which makes them not well-suited for IMTS.

## 3. Problem Definition

We consider an IMTS dataset $D := \{S_i | i = 1, ..., n\}$ consisting of $n$ samples, where $S_i$ is the $i$-th sample. With a total of $T$ timestamps and $U$ variables, each IMTS sample can be denoted as a set containing $M$ observation tuples $S_i := \{(t_j, z_j, u_j) | j = 1, ..., M\}$, where $t_j \in \{1, ..., T\}$, $z_j \in \mathbb{R}$, and $u_j \in \{1, ..., U\}$ represents the timestamp, observed value, and variable indicator respectively. For the IMTS forecasting task, given a split timestamp $t_S$, an IMTS sample is divided into a lookback window $\mathcal{X}_i := \{(t_j, z_j, u_j) | j = 1, ..., M, t_j \leq t_S\}$ and a forecast window $\mathcal{Y}_i := \{[(t_j, u_j), z_j] | j = 1, ..., M, t_j > t_S\}$. The forecast query $q_j \in \mathcal{Q}_i$ is derived by combining $(t_j, u_j)$ of the $j$-th observation tuple within the forecast window. We aim to learn a forecasting model $\mathcal{F}(\cdot)$, such that given the lookback window $\mathcal{X}_i$ and forecast query $\mathcal{Q}_i$ as input, it accurately predicts the corresponding observed values $\mathcal{Z}_i$:

$$\mathcal{F}(\mathcal{X}_i, \mathcal{Q}_i) \to \mathcal{Z}_i. \tag{1}$$

## 4. Methodology

We first detail the efficient hypergraph representation for IMTS samples in Section 4.1. Subsequently, we explain how to leverage it for the IMTS forecasting task in Section 4.2.

### 4.1. Efficient Hypergraph Representation for IMTS

Instead of using padding or patching, we convert original observations into node embeddings within the hypergraph, thereby improving efficiency in both data preprocessing and model training. As shown in Figure 2, the hypergraph is defined as $\mathcal{G} := \{\mathcal{V}, \mathcal{E}\}$. The observation node embeddings are expressed as $\mathcal{V} := \{v_j | j = 1, ..., M\}$, and two types of hyperedge embeddings $\mathcal{E} := \mathcal{E}_T \cap \mathcal{E}_U$ are defined, including temporal hyperedge embeddings $\mathcal{E}_{\text{time}} := \{e_t | t = 1, ..., T\}$ and variable hyperedge embeddings $\mathcal{E}_{\text{var}} := \{e_u | u = 1, ..., U\}$. The topology of a hypergraph can be represented using two incidence matrices: $\mathbf{H}^T \in \mathbb{R}^{M \times T}$ for temporal hyperedges, and $\mathbf{H}^U \in \mathbb{R}^{M \times U}$ for variable hyperedges. Entries $\mathbf{H}_{jt}$ in $\mathbf{H}^T$ are defined as:

$$\mathbf{H}_{jt} = \begin{cases} 1, & v_j \in e_t \\ 0, & v_j \notin e_t \end{cases}. \tag{2}$$

And entries $\mathbf{H}_{ju}$ in $\mathbf{H}^U$ are defined in the same way as Eq. (2). Unlike the edge in traditional graph structures which can only connect two nodes, a hyperedge can connect an arbitrary number of nodes, enabling hypergraphs to capture more complex relationships among nodes. Using the above notations, the IMTS forecasting problem is framed as the node prediction task in hypergraphs, where nodes representing forecast targets are the ones to predict.

We initialize node embeddings by encoding the observed values $\mathcal{Z}_i$ of the $i$-th sample through a non-linear mapping $\text{ReLU}(\text{FF}_{\text{obs}}(\cdot))$ into $P_{\text{obs}}$-dimensional embedding $\mathbf{V} \in \mathbb{R}^{M \times P_{\text{obs}}}$:

$$\mathbf{V} = \text{ReLU}(\text{FF}_{\text{obs}}(\mathcal{Z}_i)), \tag{3}$$

where the values of forecast targets are initialized to 0. For temporal hyperedge embeddings from timestamp set $T_i$, we initialize them using sinusoidal encoding after linear mapping $\text{FF}_{\text{time}}(\cdot)$ to obtain the $P_{\text{time}}$-dimensional temporal embedding $\mathbf{E}_{\text{time}} \in \mathbb{R}^{T \times P_{\text{time}}}$:

$$\mathbf{E}_{\text{time}} = \sin(\text{FF}_{\text{time}}(T_i)). \tag{4}$$

For variable hyperedge embeddings from variable set $U_i$, we use learnable parameters $\mathbf{W}_{\text{var}}$ of shape $U \times P_{\text{var}}$ for initializations, where $P_{\text{var}}$ denotes the variable embedding dimension:

$$\mathbf{E}_{\text{var}} = \text{ReLU}(\mathbf{W}_{\text{var}}). \tag{5}$$

### 4.2. Forecasting with HyperIMTS

The overview of our proposed model, HyperIMTS, is illustrated in Figure 3. It leverages the hypergraph represen-

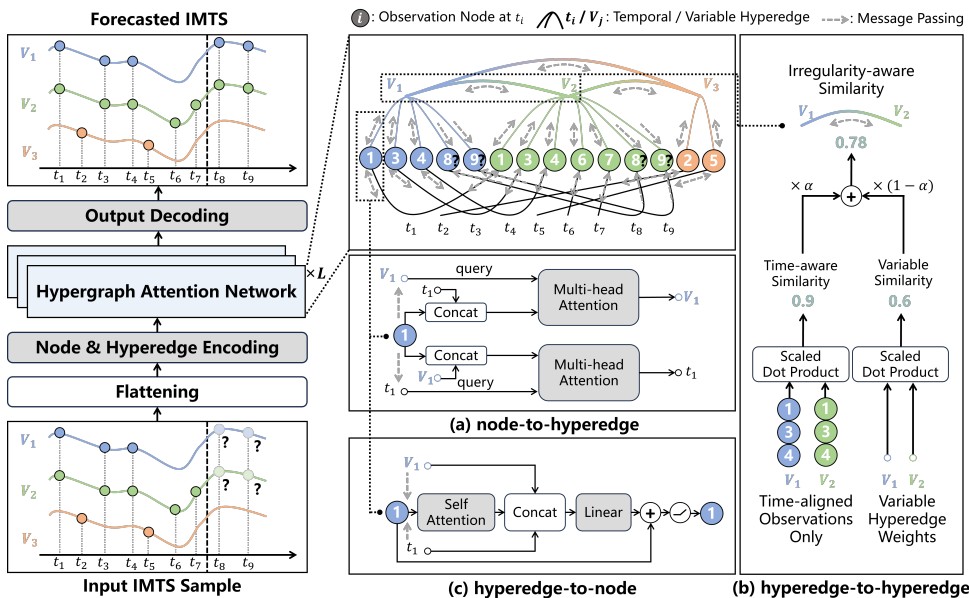

*Figure 3.* The architecture of HyperIMTS. It first converts input IMTS samples with empty forecast targets into the proposed efficient hypergraph representation. Three types of message passing are used sequentially: (a) Temporal and variable hyperedge embeddings are updated via node-to-hyperedge message passing; (b) Inter-variable correlations are modeled during irregularity-aware hyperedge-to-hyperedge message passing, where time-aware and overall variable similarities are merged based on time alignment; (c) Both temporal and variable hyperedge embeddings are used for unified hyperedge-to-node updates.

tation detailed in Section 4.1 to represent IMTS samples. Learning for temporal and variable dependencies is realized using three types of message passing in the hypergraph, where node-to-hyperedge is introduced in Section 4.2.1, hyperedge-to-hyperedge is discussed in Section 4.2.2, and hyperedge-to-node is presented in Section 4.2.3. It should be noted that temporal dependencies are learned through message passings among observations of the same variable, and variable dependencies are learned through message passings between different variables.

### 4.2.1. TEMPORAL AND VARIABLE HYPEREDGE UPDATE

In this section, we introduce node-to-hyperedge message passing for updating temporal and variable hyperedge embeddings, as illustrated in Figure 3 (a). HyperIMTS initializes temporal hyperedge embeddings with sinusoidal encoding, making them learnable to account for the varying number of observations at each timestamp. With temporal hyperedge embeddings used as multi-head attention queries $\mathbf{q}^h = \mathrm{FF}_q(\mathbf{E}_{\mathrm{time}})$ and concatenating observation node embeddings with variable hyperedge embeddings as keys $\mathbf{k}^h = \mathrm{FF}_k(\mathbf{V}||\mathbf{E}_{\mathrm{var}})$ and values $\mathbf{v}^h = \mathrm{FF}_v(\mathbf{V}||\mathbf{E}_{\mathrm{var}})$, the message passing for updated temporal hyperedge em-

beddings $\mathbf{E}'_{\mathrm{time}}$ can be expressed as:

$$\mathbf{O} = ||_{h=1}^{H}\mathrm{Softmax}(\frac{\mathbf{q}^h\mathbf{k}^{h\mathsf{T}}}{\sqrt{d/H}})\mathbf{v}^h, \qquad (6)$$

$$\mathbf{E}'_{\mathrm{time}} = \mathbf{O} + \mathrm{ReLU}(\mathrm{FF}_{\mathbf{O}}(\mathbf{O})), \qquad (7)$$

where $||_{h=1}^{H}$ denotes the concatenation of results from $H$ attention heads, and $d$ denotes the embedding dimension of $\mathbf{k}_{\mathrm{time}}$. We note that variable hyperedge embeddings are included in both keys and values to identify the corresponding variable for each observation.

Besides updating temporal hyperedge embeddings, variable hyperedge embeddings are also updated using multi-head attention. The query is defined as $\mathbf{q}_{\mathrm{var}}^h = \mathrm{FF}_q(\mathbf{E}_{\mathrm{var}})$, and timestamp information is included in the keys $\mathbf{k}_{\mathrm{var}}^h = \mathrm{FF}_k(\mathbf{V}||\mathbf{E}_{\mathrm{time}})$ and values $\mathbf{v}_{\mathrm{var}}^h = \mathrm{FF}_v(\mathbf{V}||\mathbf{E}_{\mathrm{time}})$ to distinguish relative positions. Similar to Eq. (6) and (7), we can obtain the updated variable hyperedge embeddings $\mathbf{E}'_{\mathrm{var}}$.

By updating the embeddings of temporal and variable hyperedges, messages from observations are now aggregated onto hyperedges, and we describe their further propagation in the following sections.

### 4.2.2. INTER-VARIABLE MESSAGE PASSING

In this section, we introduce the irregularity-aware hyperedge-to-hyperedge message passing among variables, as illustrated in Figure 3 (b). It begins by determining correlations among variables, which are realized through the calculation of variable similarities. We first calculate the overall variable similarities among all connected observations. For two different variables $u_a$ and $u_b$ taken from $u = 1, ..., U$, the dot-product similarity between their corresponding variable hyperedge embeddings $e_{u_a}$ and $e_{u_b}$ can be expressed as:

$$\mathbf{S}_{\text{var}} = \text{FF}_q(e_{u_a}) \cdot \text{FF}_k(e_{u_b})^{\mathsf{T}}, \tag{8}$$

where $\text{FF}_q$ and $\text{FF}_k$ are feed-forward layers for queries and keys. However, overall variable similarity is a series-level comparison, which does not leverage temporal information within the series, and may not perform optimally across all alignment scenarios. For example, similar variables $V_1$ and $V_2$ in Figure 3 (b) have 3 shared timestamps and 2 unaligned timestamps, which can lead to a positive but relatively small overall variable similarity due to the unbalanced number of observations. Based on the fact that if two time-aligned series are identical, then their time-aligned subseries are also identical, we can compare only the time-aligned observations, leading to higher similarity scores that better promote message passing. Therefore, we refine the similarity calculation by calculating time-aware similarities using time-aligned observations. Specifically, they $\mathbf{S}_{\text{obs}}$ are calculated using the dot products of connected observation nodes and can be expressed as:

$$\mathbf{S}_{\text{obs}} = [v_1^{u_a}, ..., v_{T_{\text{shared}}}^{u_a}] \cdot [v_1^{u_b}, ..., v_{T_{\text{shared}}}^{u_b}]^{\mathsf{T}}, \tag{9}$$

where $v^{u_a}$ and $v^{u_b}$ denote observation nodes for two different variables, and only $T_{\text{shared}}$ number of observations with shared timestamps across variables are selected for comparison. It should be noted that while our method and padding-based methods shown in Figure 1 (a) and (b) all leverages alignment information, we employ distinct approaches for better efficiency. Padding-based methods align IMTS by padding them before feeding them into neural networks, leading to larger data volumes to be processed across all network modules. In contrast, our method only selects time-aligned observations during hyperedge-to-hyperedge message passing, thereby eliminating the need to handle large padded data in other network modules.

We then combine two similarities based on $T_{\text{shared}}$, the total number of timestamps $T_{\text{total}}$ in two variables, and a learnable threshold parameter $\delta$ to obtain the irregular-aware variable similarity $\mathbf{S}_{\text{IMTS}}$:

$$\mathbf{S}_{\text{IMTS}} = \alpha \mathbf{S}_{\text{obs}} + (1 - \alpha) \mathbf{S}_{\text{var}}, \tag{10}$$

$$\alpha = \begin{cases} \frac{T_{\text{shared}}}{T_{\text{total}}}, & \text{for } \mathbf{S}_{\text{var}} > \delta \text{ and } \mathbf{S}_{\text{obs}} \neq 0 \\ 0, & \text{otherwise} \end{cases}, \tag{11}$$

where $\delta$ is initialized to 0.5. The logic behind $\alpha$ is to prioritize $\mathbf{S}_{\text{obs}}$ over $\mathbf{S}_{\text{var}}$ if there are more aligned observations than unaligned ones, and the variables are similar. If $\mathbf{S}_{\text{obs}} = 0$, it indicates that all observations among the variables are unaligned, such as $V_2$ and $V_3$ in Figure 3. In this case, we maintain their connection for message passing using the overall variable similarity $\mathbf{S}_{\text{var}}$.

By gathering all the similarities $\mathbf{S}_{\text{IMTS}}$ for each pair of variables as entries in the attention map $\mathbf{A}_{\text{var}}$, hyperedge-to-hyperedge message passing is implemented using scaled dot-product attention:

$$\mathbf{E}''_{\text{var}} = \text{Softmax}(\frac{\mathbf{A}_{\text{var}}}{\sqrt{d}})\text{FF}_v(\mathbf{E}'_{\text{var}}), \tag{12}$$

where $\mathbf{E}''_{\text{var}}$ represents the updated variable hyperedge embeddings after message passing, and $d$ denotes the embedding dimension.

Through irregularity-aware message passing among variable hyperedges, correlated messages are exchanged adaptive to the time alignment across variables. We discuss their further propagation to nodes in the next section.

### 4.2.3. NODE UPDATE

In this section, we introduce hyperedge-to-node message passing for node updates, as illustrated in Figure 3 (c). For the total $L$ residual layers, we use variable hyperedge embeddings $\mathbf{E}'_{\text{var}}$ without inter-variable message passing in the first $L - 1$ layers, which helps in learning temporal dependencies, also known as intra-variable dependencies. Variable hyperedge embeddings $\mathbf{E}'_{\text{var}}$ propagate messages back to connected nodes, along with timestamp information from temporal hyperedge embeddings:

$$\mathbf{V}' = \text{SelfAtten}(\mathbf{V}), \tag{13}$$

$$\mathbf{V}'' = \text{ReLU}(\mathbf{V} + \text{FF}_{\text{node}}(\mathbf{V}' || \mathbf{E}'_{\text{time}} || \mathbf{E}'_{\text{var}})), \tag{14}$$

where $\text{SelfAtten}(\mathbf{V})$ performs a self-attention update on nodes $\mathbf{V}$, and $\text{FF}_{\text{node}}$ is the linear mapping layer for nodes. At the last layer of the residual structure, we use variable hyperedge embeddings $\mathbf{E}''_{\text{var}}$ after inter-variable message passing during calculation, which helps in learning variable dependencies. It is implemented by replacing $\mathbf{E}'_{\text{var}}$ with $\mathbf{E}''_{\text{var}}$ in Eq. (14).

Through the node update process, observations receive correlated messages via temporal and variable dependencies, which also update the nodes to be predicted.

### 4.2.4. TRAINING OF HYPERIMTS

In this section, we introduce the process of obtaining forecast values and training HyperIMTS. For the $i$-th sample where $i \in n$, we convert its updated node embeddings back

*Table 1.* Experimental results on five irregular multivariate time series datasets evaluated by MSE (mean ± std). The best and second-best results are indicated in **bold** and underlined, respectively. We employ four decimal places to better demonstrate standard deviations in results.

| | Algorithm | MIMIC-III | MIMIC-IV | PhysioNet'12 | Human Activity | USHCN |
|---|---|---|---|---|---|---|
| Regular | higp | 0.9726 ± 0.0001 | 0.6616 ± 0.0001 | 0.5025 ± 0.0179 | 0.2670 ± 0.0198 | 0.2330 ± 0.0003 |
| | MOIRAI | 0.8655 ± 0.0000 | 0.4293 ± 0.0000 | 0.4924 ± 0.0000 | 0.1079 ± 0.0000 | 1.2320 ± 0.0000 |
| | FEDformer | 0.7675 ± 0.0019 | 0.8008 ± 0.0270 | 0.4150 ± 0.0002 | 0.1619 ± 0.0006 | 0.3237 ± 0.0150 |
| | Ada-MSHyper | 0.6680 ± 0.0038 | 0.4217 ± 0.0012 | 0.4143 ± 0.0008 | 0.1480 ± 0.0039 | 0.2406 ± 0.0189 |
| | Autoformer | 0.7082 ± 0.0077 | 0.5963 ± 0.0158 | 0.4137 ± 0.0040 | 0.0983 ± 0.0065 | 0.4367 ± 0.0452 |
| | TimesNet | 0.6515 ± 0.0052 | 0.4355 ± 0.0015 | 0.4076 ± 0.0006 | 0.1206 ± 0.0025 | 0.2402 ± 0.0116 |
| | iTransformer | 0.7160 ± 0.0081 | 0.5083 ± 0.0095 | 0.3975 ± 0.0023 | 0.0906 ± 0.0020 | 0.4288 ± 0.0634 |
| | FourierGNN | 0.7066 ± 0.0069 | 0.4580 ± 0.0024 | 0.3946 ± 0.0027 | 0.2822 ± 0.0100 | 0.4244 ± 0.0655 |
| | Mamba | 0.6853 ± 0.0014 | 0.6102 ± 0.0051 | 0.3906 ± 0.0012 | 0.1253 ± 0.0006 | 0.2100 ± 0.0026 |
| | TSMixer | 0.6127 ± 0.0011 | 0.3566 ± 0.0021 | 0.3817 ± 0.0007 | 0.1635 ± 0.0028 | 0.2152 ± 0.0144 |
| | PatchTST | 0.6403 ± 0.0019 | 0.2939 ± 0.0009 | 0.3781 ± 0.0009 | 0.0763 ± 0.0001 | 0.2830 ± 0.0754 |
| | Leddam | 0.5935 ± 0.0054 | 0.3697 ± 0.0019 | 0.3754 ± 0.0025 | 0.0913 ± 0.0006 | 0.2887 ± 0.0259 |
| | BigST | 0.5855 ± 0.0008 | 0.3579 ± 0.0023 | 0.3425 ± 0.0010 | 0.1671 ± 0.0147 | 0.2019 ± 0.0014 |
| | Reformer | 0.5677 ± 0.0007 | 0.3894 ± 0.0023 | 0.3573 ± 0.0003 | 0.0982 ± 0.0013 | 0.2058 ± 0.0042 |
| | Informer | 0.5657 ± 0.0037 | 0.4477 ± 0.0067 | 0.3441 ± 0.0006 | 0.0708 ± 0.0003 | 0.2000 ± 0.0014 |
| | Crossformer | 0.6136 ± 0.0128 | 0.3696 ± 0.0025 | 0.3385 ± 0.0053 | 0.1410 ± 0.0208 | 0.2121 ± 0.0035 |
| Irregular | PrimeNet | 0.9900 ± 0.0000 | 0.6650 ± 0.0000 | 0.8035 ± 0.0000 | 4.3697 ± 0.0005 | 0.4925 ± 0.0012 |
| | SeFT | 0.9916 ± 0.0001 | 0.6713 ± 0.0002 | 0.7777 ± 0.0003 | 1.4141 ± 0.0023 | 0.3345 ± 0.0010 |
| | mTAN | 0.8963 ± 0.0227 | 0.5346 ± 0.0127 | 0.3889 ± 0.0026 | 0.0925 ± 0.0020 | 0.1929 ± 0.0020 |
| | NeuralFlows | 0.7167 ± 0.0025 | 0.4737 ± 0.0018 | 0.4196 ± 0.0016 | 0.1680 ± 0.0033 | 0.2007 ± 0.0043 |
| | CRU | 0.7065 ± 0.0028 | 0.4346 ± 0.0022 | 0.6189 ± 0.0012 | 0.1374 ± 0.0040 | 0.2255 ± 0.0086 |
| | GNeuralFlow | 0.6950 ± 0.0046 | 0.5005 ± 0.0021 | 0.3881 ± 0.0032 | 0.1734 ± 0.0012 | 0.1832 ± 0.0025 |
| | GRU-D | 0.6125 ± 0.0281 | 0.6622 ± 0.0018 | 0.3462 ± 0.0004 | 0.1761 ± 0.0228 | 0.1639 ± 0.0026 |
| | Raindrop | 0.5924 ± 0.0013 | 0.3413 ± 0.0046 | 0.3918 ± 0.0014 | 0.0958 ± 0.0039 | 0.2131 ± 0.0067 |
| | tPatchGNN | 0.5173 ± 0.0037 | 0.2744 ± 0.0022 | 0.3221 ± 0.0017 | 0.0443 ± 0.0005 | 0.2010 ± 0.0191 |
| | Warpformer | 0.4869 ± 0.0007 | 0.2742 ± 0.0023 | 0.3084 ± 0.0007 | 0.0539 ± 0.0007 | **0.1565 ± 0.0012** |
| | GraFITi | 0.4534 ± 0.0015 | 0.2454 ± 0.0006 | 0.3060 ± 0.0009 | 0.0435 ± 0.0001 | 0.2026 ± 0.0107 |
| | **HyperIMTS (Ours)** | **0.4259 ± 0.0021** | **0.2174 ± 0.0009** | **0.2996 ± 0.0003** | **0.0421 ± 0.0021** | 0.1738 ± 0.0078 |

to IMTS values via output linear mapping $\text{FF}_{\text{out}}(\cdot)$:

$$\hat{\mathcal{Z}}_i = \text{FF}_{\text{out}}(\mathbf{V}''||\mathbf{E}'_{\text{time}}||\mathbf{E}''_{\text{var}}), \tag{15}$$

where each node is decoded together with the information from its connected hyperedges. The model is trained by minimizing the Mean Squared Error (MSE) loss between the prediction $\hat{\mathcal{Z}}_i$ for forecast queries and the corresponding ground truth $\mathcal{Z}_i$.

### 4.2.5. COMPUTATIONAL COMPLEXITY

The computational complexity of HyperIMTS mainly arises from three attention calculations, including multi-head attention in Eq. (6), scaled dot-product attention in Eq. (12), and self-attention in Eq. (14). For a query matrix of shape $(N_q, P)$ and a key matrix of shape $(N_k, P)$, the linear mapping complexity is $\mathcal{O}(N_q P^2)$ for the query and $\mathcal{O}(N_k P^2)$ for the key. For the dot-product operation, the computational complexity is $\mathcal{O}(N_q N_k P)$, and multiplying resultant with a value matrix of shape $(N_k, P)$ also has a complexity of $\mathcal{O}(N_q N_k P)$. During node-to-hyperedge message passing as defined in Eq. (6), $N_q$ is $U$ for variable hyperedge embeddings and $T$ for temporal ones, and $N_k$ is $M$. In the hyperedge-to-hyperedge message passing described in Eq. (12), both $N_q$ and $N_k$ are $U$ for variable hyperedges. For self attention in Eq. (14), both $N_q$ and $N_k$ are $M$ for observation nodes. Although attention is not a computa-

*Table 2.* Summary of five datasets. Canonical padding approach significantly increases number of observations, and patch-aligned padding can either be better or worse than canonical one. The patch lengths for the five datasets are 12 hours for MIMIC-III, MIMIC-IV, and PhysioNet'12, 300 milliseconds for Human Activity, and 0.2 year for USHCN, respectively.

| Description | MIMIC-III | MIMIC-IV | PhysioNet'12 | Human Activity | USHCN |
|---|---|---|---|---|---|
| Max length | 96 | 971 | 47 | 131 | 337 |
| # Variable | 96 | 100 | 36 | 12 | 5 |
| # Sample | 21,250 | 17,874 | 11,981 | 1,359 | 1,114 |
| Avg # obs. | 144.6 | 304.8 | 308.6 | 362.2 | 313.5 |
| Avg # obs. (padding) | 9,216.0 | 92,000.0 | 1,692.0 | 1,573.2 | 1,685.0 |
| Avg # obs. (patching) | 9,210.0 | 33,761.3 | 1,800.0 | 1,803.0 | 1,112.9 |

tionally efficient operation theoretically, HyperIMTS only handles observed values, resulting in smaller data volumn compared to padding approaches. The statistics of actual datasets can be found in Table 2.

## 5. Experiments

### 5.1. Experimental Setup

#### 5.1.1. DATASETS

Five widely studied irregular multivariate time series datasets, covering healthcare, biomechanics, and climate, are used in the experiments, and their statistics are summarized in Table 2. MIMIC-III (Johnson et al., 2016) is a

*Table 3.* Ablation results of HyperIMTS and its five variants on five irregular multivariate time series datasets.

| Ablation | MIMIC-III | MIMIC-IV | PhysioNet'12 | Human Activity | USHCN |
|---|---|---|---|---|---|
| Complete | **0.4259 ± 0.0021** | **0.2174 ± 0.0009** | **0.2996 ± 0.0003** | **0.0421 ± 0.0021** | **0.1738 ± 0.0078** |
| w/o VE | 0.9556 ± 0.0029 | 0.6293 ± 0.0053 | 0.6945 ± 0.0002 | 1.3855 ± 0.0006 | 0.3299 ± 0.0054 |
| w/o IAVD | 0.4466 ± 0.0007 | 0.2358 ± 0.0012 | 0.3050 ± 0.0006 | 0.0481 ± 0.0004 | 0.1983 ± 0.0105 |
| rp IAVD | 0.4317 ± 0.0030 | 0.2189 ± 0.0024 | 0.3014 ± 0.0003 | 0.0463 ± 0.0008 | 0.1794 ± 0.0062 |
| w/o TE | 0.4954 ± 0.0043 | 0.2652 ± 0.0005 | 0.3029 ± 0.0003 | 0.0745 ± 0.0001 | 0.1757 ± 0.0088 |
| rp TE | 0.4403 ± 0.0024 | 0.2333 ± 0.0004 | 0.3029 ± 0.0003 | 0.0473 ± 0.0007 | 0.1894 ± 0.0118 |

clinical database collected from ICU patients during the first 48 hours of admission, which is rounded for 30 minutes. MIMIC-IV (Johnson et al., 2023) is built upon MIMIC-III, which has higher sampling frequency and rounded for 1 minute. PhysioNet'12 (Silva et al., 2012) is also a clinical database collected from the first 48 hours of ICU stay, which is rounded for 1 hour. Human Activity contains biomechanics data describing 3D positional variables, which is rounded for 1 millisecond. USHCN (Menne et al., 2016) includes climate data over 150 years collected by meteorological stations scattered over the United States, and we focus on a subset of 4 years between 1996 and 2000. For PhysioNet'12, MIMIC-III, MIMIC-IV, and USHCN, we follow the preprocessing setup in previous work (Yalavarthi et al., 2024). For Human Activity, we follow the preprocessing set up in (Zhang et al., 2024). All five datasets are split into training, validation, and test sets adhering to ratios of 80%, 10%, and 10%, respectively. Training and validation sets are shuffled, while test sets are not.

### 5.1.2. BASELINES

To conduct comprehensive and fair comparisons, we design a unified and extensible pipeline to evaluate models across various domains and tasks. Twenty-seven baselines are included in the benchmark, covering SOTA methods from (1) Multivariate time series forecasting: FEDformer (Zhou et al., 2022), Ada-MSHyper (Shang et al., 2024), Autoformer (Wu et al., 2021), TimesNet (Wu et al., 2022), iTransformer (Liu et al., 2023), FourierGNN (Yi et al., 2023), Mamba (Gu & Dao, 2024), TSMixer (Chen et al., 2023), PatchTST (Nie et al., 2022), Leddam (Yu et al., 2024), Reformer (Kitaev et al., 2019), Informer (Zhou et al., 2021), Crossformer (Zhang & Yan, 2022), (2) Time series pretraining: MOIRAI (Woo et al., 2024), PrimeNet (Chowdhury et al., 2023), (3) Traffic forecasting: higp (Cini et al., 2024), BigST (Han et al., 2024), (4) IMTS classification, imputation, and forecasting: SeFT (Horn et al., 2020), mTAN (Shukla & Marlin, 2020), NeuralFlows (Biloš et al., 2021), CRU (Schirmer et al., 2022), GNeuralFlow (Mercatali et al., 2024), GRU-D (Che et al., 2018), Raindrop (Zhang et al., 2021), tPatchGNN (Zhang et al., 2024), Warpformer (Zhang et al., 2023a), GraFITi (Yalavarthi et al.,

2024). Further details on introductions and hyperparameter settings of these baselines can be found in Appendix A.4.

### 5.1.3. IMPLEMENTATION DETAILS

All models are trained on a Linux server with PyTorch version 2.4.1 using two NVIDIA GeForce RTX 3090 GPUs, and efficiency analysis is conducted on another Linux server with PyTorch version 2.2.2+cu118 using one NVIDIA GeForce RTX 2080Ti GPU. The learning rate $\mathcal{L}_n$ for the $n$-th epoch is kept unchanged when $n <= 3$, and adjusted according to $\mathcal{L}_n = \mathcal{L}_0 \times 0.8^{n-3}$ when $n > 3$, where the initial learning rate $\mathcal{L}_0$ is specific to each model and dataset and can be found in Appendix A.4. All experiments run with a maximum epoch number of 300 and early stopping patience of 10 epochs. To mitigate randomness, we conduct each experiment using five different random seeds ranging from 2024 to 2028 and calculate the mean and standard deviation of the results. MSE is used as the training loss function for models, unless a custom loss function proposed in the original paper is used. When adapting regular time series models for IMTS, masks indicating observed values are included in MSE calculations during training. The detailed settings for the hyperparameters are provided in Appendix A.4. We note that our experimental results may differ from those in existing papers, mainly due to differences in normalization methods, random seeds, and learning rate schedulers. We eliminate these differences to ensure fair comparisons.

### 5.2. Main Results

Table 1 shows the models' forecasting performance, evaluated using MSE on five datasets, where the best results are highlighted in bold and the next best shown in underline. The lookback time periods are 36 hours for MIMIC-III, MIMIC-VI, and PhysioNet'12, 3000 milliseconds for Human Activity, and 3 years for USHCN. Human Activity use 300 milliseconds as forecast length, and the rest datasets use the next 3 timestamps as forecast targets, following the settings in existing works (Biloš et al., 2021; De Brouwer et al., 2019). Results evaluated using MAE are detailed in Appendix A.1, and an analysis of varying lookback and forecast horizons can be found in Appendix A.2. As can be

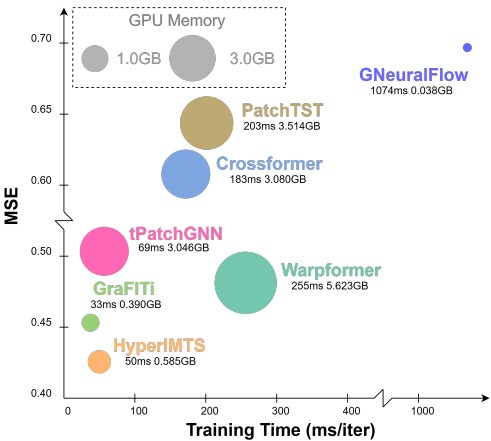

*Figure 4.* Model efficiency comparison on MIMIC-III, with 36 hours of lookback length, 3 forecast timestamps, 96 variables, and a batch size of 32. Our proposed model, HyperIMTS, achieves the lowest MSE while maintaining high computational efficiency, as measured by training time and memory footprints.

seen, HyperIMTS excels as the best-performing model in four out of the five datasets. The results on USHCN exhibit high variance, as noted in previous work (Yalavarthi et al., 2024). However, these results are also provided for comprehensive comparisons. By leveraging irregularity-aware message passing in hypergraphs for dependency learning on all observations, HyperIMTS provides up to 11.4% improvement in comparison with the overall next best model GraFITi. On average, HyperIMTS exhibits significant reduction in MSE compared to the best-performing regular time series model, Crossformer. We observed that some models designed for regular time series outperform a few IMTS models, highlighting the necessity of evaluating both regular and irregular time series models for more comprehensive comparisons. Also, the time series pretraining models MOIRAI and PrimeNet do not perform particularly well in general. This suggests potential discrepancies among different datasets, particularly between regular and irregular time series datasets, which requires further efforts in designing the pretraining process.

### 5.3. Ablation Study

We evaluate the performance of HyperIMTS and five of its variants on all five datasets. (1) **rp TE** replaces learnable temporal hyperedges with non-learnable sinusoidal embeddings; (2) **w/o TE** removes all temporal hyperedges; (3) **rp IAVD** replaces irregularity-aware variable dependencies with overall variable dependencies only; (4) **w/o IAVD** removes irregular-aware variable dependencies by disabling hyperedge-to-hyperedge message passing among variable hyperedges; (5) **w/o VE**: removes all variable hyperedges;

The ablation results are summarized in Table 3. As can be seen, all model components are necessary. Results from **w/o IAVD** and **rp IAVD** show the necessity of accounting for variable dependencies when dealing with irregular observation timestamps. **rp TE** demonstrates the importance of learnable temporal representations in effectively leveraging timestamp information within IMTS. **w/o TE** and **w/o VE** show the effectiveness of both temporal and variable hyperedges for feature extraction and message passing.

### 5.4. Efficiency Analysis

We select the most competitive baselines as well as representative methods for the efficiency comparison. From a data preprocessing perspective, these methods can be categorized as follows: (1) Non-padding methods: HyperIMTS and GraFITi; (2) Patch-aligned padding method: tPatchGNN; (3) Canonical padding methods: Warpformer and GNeuralFlow; (4) Fully observed MTS methods: Crossformer and PatchTST. Models are evaluated based on their MSE, training time, and GPU memory footprints. The training time for one epoch with a batch size of 32 is recorded, then divided by the number of batches to determine the training time per iteration. Memory footprints only encompass the model's usage instead of representing the entire process. The results on MIMIC-III are shown in Figure 4. As can be seen, our model HyperIMTS achieves the lowest MSE while keeping computational costs relatively low. It can also be observed that non-padding models like HyperIMTS and GraFITi achieve faster training speeds compared to other models. tPatchGNN uses patch-aligned padding, which can reduces the average number of padding values in MIMIC-III and runs faster than models using canonical padding. However, it still consumes a considerable amount of GPU memory, and may result in more padding values than canonical approaches for samples with numerous asynchronous observations. Transformer-based models, including Warpformer, Crossformer, and PatchTST, also exhibit large memory usage. This is primarily due to the attention calculations on padded samples, which involve a greater amount of data compared to original samples. Although using a small amount of memory, the ODE-based model GNeuralFlow takes significantly longer training time compared to other models, indicating the potential inefficiency of ODE solvers. Further efficiency analysis for varying lookback lengths as well as results on other datasets are available in Appendix A.3

### 6. Conclusion

This paper introduces a hypergraph neural network approach, HyperIMTS, to address the IMTS forecasting problem. HyperIMTS represents observed values as nodes in the hypergraph without padding, and connects them with

hyperedges denoting timestamps and variables. Through three types of message passing in hypergraphs, HyperIMTS effectively captures temporal and variable dependencies in a unified and efficient manner. Moreover, by leveraging time-aware similarity from observation nodes and overall similarity from variable hyperedges, HyperIMTS adaptively choose the optimal way to model variable correlations based on time alignment. HyperIMTS demonstrates competitive performance on the IMTS forecasting task across twenty-seven state-of-the-art time series models in our unified code pipeline. Nevertheless, there are still limitations for Hyper-IMTS. Our model currently does not support multi-modal data, such as text notes or images, which are found in medical IMTS datasets and may enhance forecasting. Additionally, attention calculations are more resource-intensive compared to other recent methods, such as state space approaches. We will address these limitations in future work.

## Acknowledgements

We thank the anonymous reviewers for their helpful feedbacks, and all the donors of the original datasets. The work described in this paper was partially funded by the National Natural Science Foundation of China (Grant Nos. 62272173), the Natural Science Foundation of Guangdong Province (Grant Nos. 2024A1515010089, 2022A1515010179), the Science and Technology Planning Project of Guangdong Province (Grant No. 2023A0505050106), and the National Key R&D Program of China (Grant No. 2023YFA1011601).

## Impact Statement

This paper details efforts to advance irregular multivariate time series forecasting across various scientific domains. While our research may have broader societal impacts, we do not consider it necessary to single out any particular consequences for emphasis here.

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

*Table 4.* Experimental results on five irregular multivariate time series datasets, evaluated using MAE. The experimental setup is the same as Table 1.

| | Algorithm | MIMIC-III | MIMIC-IV | PhysioNet'12 | Human Activity | USHCN |
|---|---|---|---|---|---|---|
| Regular | higp | 0.6529 ± 0.0003 | 0.5804 ± 0.0001 | 0.5183 ± 0.0140 | 0.3870 ± 0.0115 | 0.3411 ± 0.0003 |
| | MOIRAI | 0.5790 ± 0.0000 | 0.3998 ± 0.0000 | 0.4937 ± 0.0000 | 0.1914 ± 0.0000 | 0.8274 ± 0.0000 |
| | FEDformer | 0.5651 ± 0.0006 | 0.6469 ± 0.0157 | 0.4549 ± 0.0003 | 0.2897 ± 0.0009 | 0.3704 ± 0.0060 |
| | Ada-MSHyper | 0.5354 ± 0.0024 | 0.4423 ± 0.0008 | 0.4579 ± 0.0037 | 0.2610 ± 0.0037 | 0.3186 ± 0.0074 |
| | Autoformer | 0.5475 ± 0.0033 | 0.5477 ± 0.0096 | 0.4534 ± 0.0031 | 0.2178 ± 0.0062 | 0.4102 ± 0.0047 |
| | TimesNet | 0.5232 ± 0.0012 | 0.4511 ± 0.0005 | 0.4538 ± 0.0005 | 0.2258 ± 0.0023 | 0.3378 ± 0.0021 |
| | iTransformer | 0.5581 ± 0.0052 | 0.4875 ± 0.0068 | 0.4443 ± 0.0019 | 0.2018 ± 0.0024 | 0.3365 ± 0.0065 |
| | FourierGNN | 0.5381 ± 0.0021 | 0.4499 ± 0.0016 | 0.4386 ± 0.0019 | 0.3764 ± 0.0069 | 0.4013 ± 0.0206 |
| | Mamba | 0.5423 ± 0.0010 | 0.5612 ± 0.0034 | 0.4390 ± 0.0008 | 0.2330 ± 0.0006 | 0.3245 ± 0.0025 |
| | TSMixer | 0.4949 ± 0.0003 | 0.3977 ± 0.0011 | 0.4351 ± 0.0004 | 0.2836 ± 0.0037 | 0.3010 ± 0.0033 |
| | PatchTST | 0.5153 ± 0.0011 | 0.3246 ± 0.0006 | 0.4325 ± 0.0005 | 0.1723 ± 0.0014 | 0.3502 ± 0.0424 |
| | Leddam | 0.4828 ± 0.0033 | 0.3985 ± 0.0018 | 0.4281 ± 0.0022 | 0.2003 ± 0.0008 | 0.3134 ± 0.0073 |
| | BigST | 0.4734 ± 0.0004 | 0.3990 ± 0.0015 | 0.4012 ± 0.0005 | 0.2695 ± 0.0108 | 0.3322 ± 0.0029 |
| | Reformer | 0.4724 ± 0.0004 | 0.4253 ± 0.0014 | 0.4117 ± 0.0002 | 0.2282 ± 0.0022 | 0.3136 ± 0.0042 |
| | Informer | 0.4682 ± 0.0026 | 0.4592 ± 0.0044 | 0.4059 ± 0.0002 | 0.1833 ± 0.0010 | 0.3186 ± 0.0024 |
| | Crossformer | 0.4961 ± 0.0092 | 0.4021 ± 0.0022 | 0.3954 ± 0.0054 | 0.2498 ± 0.0170 | 0.3322 ± 0.0058 |
| Irregular | PrimeNet | 0.6690 ± 0.0000 | 0.5879 ± 0.0001 | 0.6887 ± 0.0000 | 1.7268 ± 0.0001 | 0.4945 ± 0.0007 |
| | SeFT | 0.6661 ± 0.0004 | 0.5895 ± 0.0005 | 0.6742 ± 0.0003 | 0.9923 ± 0.0006 | 0.4079 ± 0.0038 |
| | mTAN | 0.6437 ± 0.0088 | 0.5184 ± 0.0057 | 0.4390 ± 0.0019 | 0.2181 ± 0.0032 | 0.3335 ± 0.0019 |
| | NeuralFlows | 0.5490 ± 0.0013 | 0.4794 ± 0.0010 | 0.4604 ± 0.0014 | 0.3088 ± 0.0035 | 0.3144 ± 0.0032 |
| | CRU | 0.5369 ± 0.0014 | 0.4558 ± 0.0011 | 0.5815 ± 0.0008 | 0.2572 ± 0.0040 | 0.3371 ± 0.0074 |
| | GNeuralFlow | 0.5352 ± 0.0027 | 0.4899 ± 0.0003 | 0.4377 ± 0.0026 | 0.3146 ± 0.0016 | 0.2980 ± 0.0034 |
| | GRU-D | 0.4891 ± 0.0138 | 0.5816 ± 0.0012 | 0.4065 ± 0.0003 | 0.3155 ± 0.0205 | 0.2923 ± 0.0019 |
| | Raindrop | 0.4850 ± 0.0008 | 0.3879 ± 0.0032 | 0.4410 ± 0.0012 | 0.2174 ± 0.0046 | 0.3128 ± 0.0076 |
| | tPatchGNN | 0.4293 ± 0.0039 | 0.3096 ± 0.0016 | 0.3825 ± 0.0024 | 0.1238 ± 0.0003 | 0.2928 ± 0.0068 |
| | Warpformer | 0.4039 ± 0.0009 | 0.3142 ± 0.0022 | 0.3667 ± 0.0009 | 0.1296 ± 0.0013 | **0.2704 ± 0.0020** |
| | GraFITi | 0.3923 ± 0.0010 | 0.3004 ± 0.0004 | 0.3621 ± 0.0007 | 0.1204 ± 0.0006 | 0.3029 ± 0.0145 |
| | **HyperIMTS (Ours)** | **0.3800 ± 0.0009** | **0.2837 ± 0.0007** | **0.3598 ± 0.0002** | **0.1199 ± 0.0059** | 0.2773 ± 0.0064 |

# A. Additional Experiments

## A.1. Different Metrics

We present the results measured using MAE in Table 4, which follows the same experimental setup as Table 1. As can be seen, HyperIMTS outperforms twenty-seven baselines on four of the five datasets and comes in second place on the remaining one. The variance in results for the USHCN dataset remains high, similar to the findings from MSE evaluations, as discussed in Section 5.2.

## A.2. Varying Lookback Lengths and Forecast Horizons

We also assess performance across varying lookback and forecast horizons. For varying forecast horizons, we follow the same settings in previous work (Zhang et al., 2024) and keep the lookback length settings the same as in Table 1. For MIMIC-III, MIMIC-IV, and PhysioNet'12, the forecast horizon is set to 12 hours. For Human Activity, the forecast horizon is 1000 milliseconds. For USHCN, the forecast horizon is set to 1 year. The results are summarized in Table 5, where we select the most competitive baselines from Table 1 and include some well-known models for comparison, including five MTS forecasting models and five IMTS models. It can be seen that HyperIMTS remains the overall best-performing model evaluated based on ranking. We note that, unlike fully observed MTS, IMTS samples are typically split based on time periods rather than the number of observations. The forecast settings here view 75% of the time period as the lookback window, while the remaining 25% as the forecast window.

For varying lookback lengths, the results are shown in Figure 5. We keep the forecast horizons the same as in Table 5, and varying the lookback lengths to: (1) MIMIC-III: 12, 24, and 36 hours; (2) MIMIC-IV: 12, 24, and 36 hours; (3) PhysioNet'12: 12, 24, and 36 hours; (4) Human Activity: 1000, 2000, and 3000 milliseconds; (5) USHCN: 1, 2, and 3 years. As can be seen in the result, the forecasting performance of HyperIMTS generally improves with increasing lookback length, except for MIMIC-III, where all models perform worse at 36 hours than at 24 hours. It is possible that the change in temporal patterns, observed after a 24-hour stay for patients, may be responsible.

*Table 5.* Experimental results on five irregular multivariate time series datasets evaluated using MSE, with the lookback length following Table 1 and forecast horizons set to the rest length of the whole series, which are 12 hours for MIMIC-III, MIMIC-IV, and PhysioNet'12, 1000 milliseconds for Human Activity, and 1 year for USHCN.

| | Algorithm | MIMIC-III | MIMIC-IV | PhysioNet'12 | Human Activity | USHCN |
|---|---|---|---|---|---|---|
| Regular | Ada-MSHyper | $0.7437 \pm 0.0098$ | $0.4305 \pm 0.0035$ | $0.4593 \pm 0.0006$ | $0.1627 \pm 0.0027$ | $0.5001 \pm 0.0047$ |
| | iTransformer | $0.7825 \pm 0.0061$ | $0.4815 \pm 0.0014$ | $0.4545 \pm 0.0020$ | $0.0997 \pm 0.0025$ | $0.5624 \pm 0.0051$ |
| | PatchTST | $0.6976 \pm 0.0008$ | $0.3704 \pm 0.0010$ | $0.4374 \pm 0.0005$ | $0.0868 \pm 0.0006$ | $0.6816 \pm 0.1005$ |
| | Informer | $0.6606 \pm 0.0063$ | $0.3752 \pm 0.0025$ | $0.4050 \pm 0.0004$ | $0.0837 \pm 0.0003$ | $\mathbf{0.4337 \pm 0.0013}$ |
| | Crossformer | $0.6412 \pm 0.0018$ | $0.3624 \pm 0.0022$ | $0.4080 \pm 0.0022$ | $0.1656 \pm 0.0289$ | $0.4672 \pm 0.0092$ |
| Irregular | GNeuralFlow | $0.7434 \pm 0.0068$ | $0.4826 \pm 0.0022$ | $0.4390 \pm 0.0022$ | $0.2074 \pm 0.0281$ | $0.4988 \pm 0.0013$ |
| | GRU-D | $0.6367 \pm 0.0020$ | $0.4446 \pm 0.0005$ | $0.4059 \pm 0.0006$ | $0.1829 \pm 0.0229$ | $0.5320 \pm 0.0056$ |
| | tPatchGNN | $0.5896 \pm 0.0055$ | $0.2881 \pm 0.0008$ | $0.3806 \pm 0.0009$ | $0.0601 \pm 0.0005$ | $0.5709 \pm 0.0438$ |
| | Warpformer | $0.5664 \pm 0.0020$ | $0.2991 \pm 0.0020$ | $0.3730 \pm 0.0008$ | $0.0611 \pm 0.0010$ | $0.4531 \pm 0.0005$ |
| | GraFITi | $\underline{0.5348 \pm 0.0008}$ | $\underline{0.2722 \pm 0.0008}$ | $\underline{0.3772 \pm 0.0001}$ | $\underline{0.0596 \pm 0.0007}$ | $0.4419 \pm 0.0092$ |
| | **HyperIMTS (Ours)** | $\mathbf{0.5111 \pm 0.0015}$ | $\mathbf{0.2328 \pm 0.0005}$ | $\mathbf{0.3683 \pm 0.0002}$ | $\mathbf{0.0589 \pm 0.0020}$ | $\underline{0.4400 \pm 0.0049}$ |

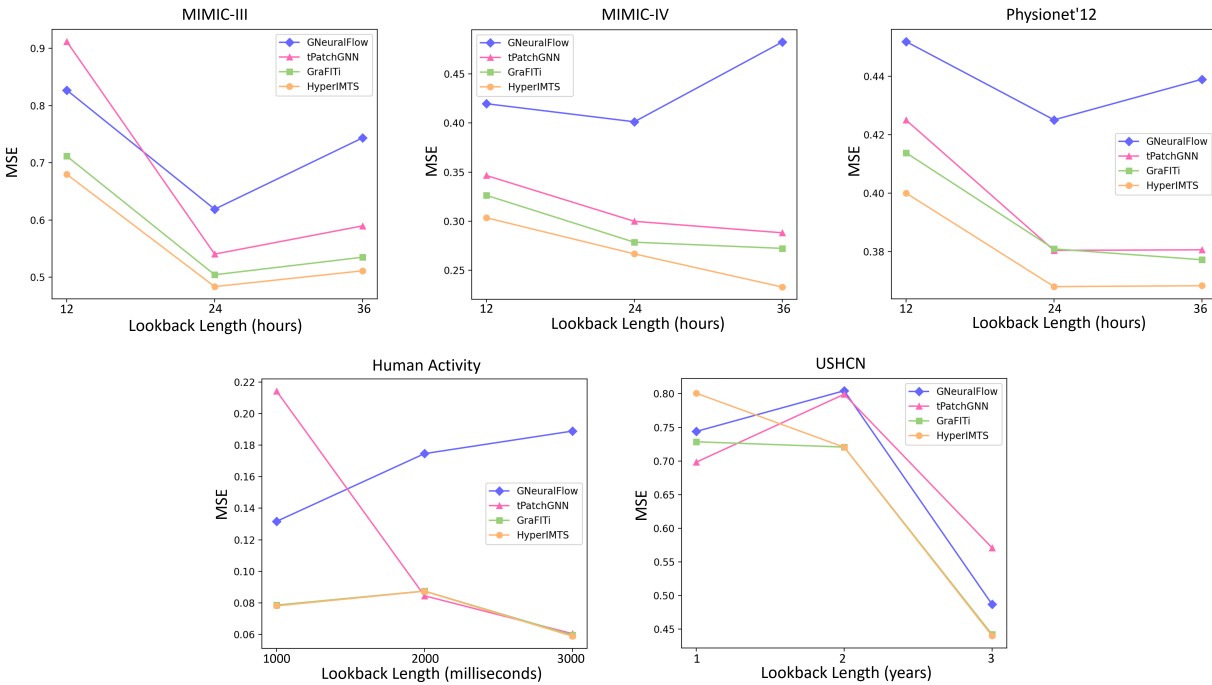

*Figure 5.* Forecasting performance with varying lookback lengths and fixed forecast horizons.

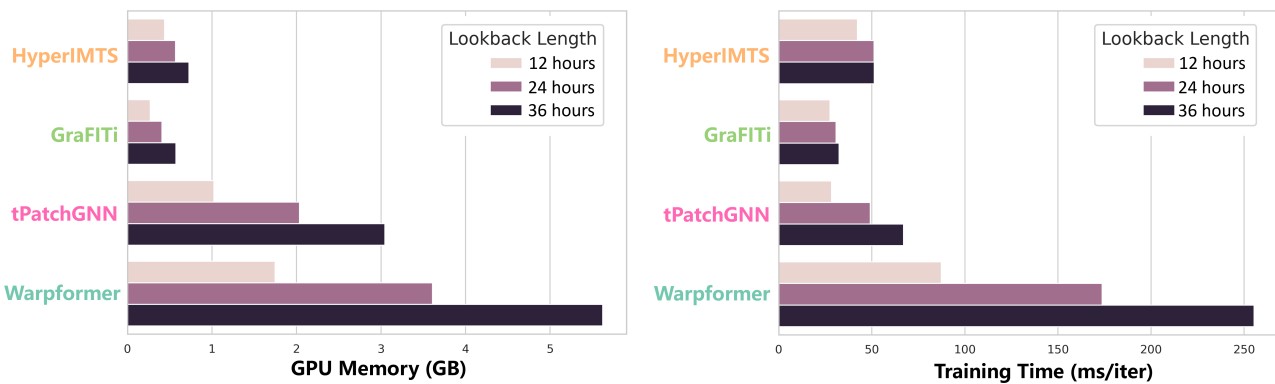

*Figure 6.* Efficiency comparison with varying lookback lengths and fixed forecast horizons on MIMIC-III. The efficiency of non-padding methods is relatively insensitive to the increase in lookback length.

### A.3. Additional Efficiency Analysis

We further analyze the efficiency of non-padding, patch-aligned padding, and canonical padding approaches for different lookback lengths. We fix the forecast horizons to 12 hours in MIMIC-III, and vary the lookback lengths from 12 to 36 hours. The results are summarized in Figure 6, where HyperIMTS and GraFITi are non-padding approaches, tPatchGNN uses patch-aligned padding, and Warpformer as well as GNeuralFlow are canonical padding approaches. As can be seen, efficiency for non-padding methods is relatively insensitive to increases in lookback length. Patch-aligned padding method tPatchGNN also achieves relatively fast speed, but deteriorates quickly with the increase of lookback length. Non-padding method Warpformer spends the highest amount of GPU memory as well as training time compared to others.

We also provide four additional efficiency comparisons on MIMIC-IV, PhysioNet'12, Human Activity, and USHCN using the same settings as Table 1, and the results are summarized in Figure 7. They confirm the finding that efficiency of non-padding methods is relatively insensitive to increases in lookback length, as MIMIC-IV and USHCN have larger max time length than PhysioNet'12 and Human Activity. When the data volumn is small, non-padding methods have a similar base cost to other approaches, explaining their slower performance on PhysioNet'12 and Human Activity. We also notice the unexpected efficiency of Crossformer on MIMIC-IV. This may be attributed to the router mechanism in the cross-dimensional stage of Crossformer, as MIMIC-IV has the largest number of variables among the five IMTS datasets.

In summary, HyperIMTS demonstrates superior overall performance while maintaining low computational costs across all datasets and input settings. Non-padding approaches exhibit a similar base cost to other padding methods when the lookback length is relatively small. However, their efficiency is insensitive to the increase in lookback length. Therefore, they can generally use less memory and training time for longer inputs. Further improvements on non-padding methods can focus on more efficient dependency modeling mechanisms, such as the router mechanism used in Crossformer.

### A.4. Baseline Details

We briefly introduce each baseline model along with their key hyperparameter settings here. Unless otherwise specified, we use a batch size of 16 for USHCN, and 32 for others. We try to use the same hyperparameter settings for learning rate, hidden dimension, special loss functions, and number of layers from their original papers and codes, if available. Number of epochs, early stopping patience, random seeds, and learning rates have been described in Section 5.1.3. For all classification models, we replace the final softmax layer with a linear layer to enable forecasting.

#### A.4.1. METHODS FOR MTS

**higp** (Cini et al., 2024) uses hierarchical structure along variable dimension for traffic forecasting. The hidden dimension is 512. The learning rate is $1 \times 10^{-3}$. Since the model requires pre-defined variable adjacency matrix in dataset, which is not available for IMTS datasets, we borrow the idea from graph neural networks and replace it with graph learner in the model.

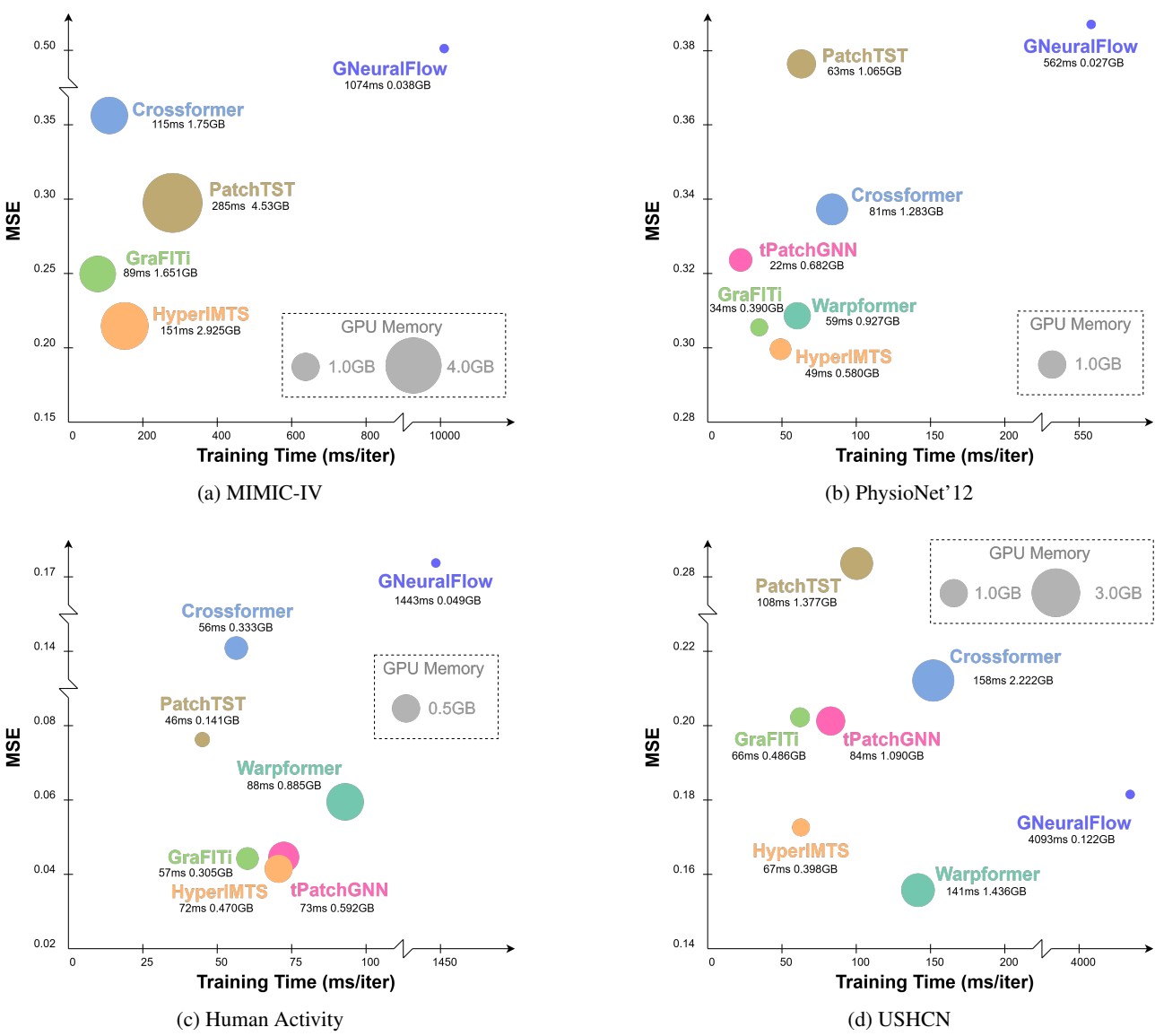

*Figure 7.* Efficiency comparisons on MIMIC-IV, PhysioNet'12, Human Activity, and USHCN. tPatchGNN and Warpformer run out of GPU memory on MIMIC-IV and are not plotted. Efficiency for non-padding methods is similar to other approaches on shorter lookback length like PhysioNet'12 and Human Activity, but they are less sensitive to the grouth of length, thus faster than most other methods on MIMIC-IV and USHCN.

**MOIRAI** (Woo et al., 2024) is a pretraining model for time series forecasting. We use the small version of the provided pretrained configurations and weights, comprising 6 layers with hidden dimension 384. We finetune the model on IMTS datasets with learning rate of $1 \times 10^{-4}$.

**FEDformer** (Zhou et al., 2022) is a frequency-enhanced decomposed transformer for MTS forecasting. The attention factor is 3. The hidden dimension is 512. The learning rate is $1 \times 10^{-4}$.

**Ada-MSHyper** (Shang et al., 2024) uses hypergraphs for temporal multiscale learning in MTS forecasting. The window size for multiscale is 4. The learning rate is $1 \times 10^{-3}$. We also use its node and hyperedge constrainted loss function for training.

**Autoformer** (Wu et al., 2021) is a transformer variant with auto-correlation decomposition for MTS forecasting. The attention factor is 3. The hidden dimension is 512. The learning rate is $1 \times 10^{-3}$.

**TimesNet** (Wu et al., 2022) uses 2-D variation modeling for MTS analysis. The attention factor is 3. The hidden dimension is 512. The dimension for FCN is 32. The learning rate is $1 \times 10^{-3}$.

**iTransformer** (Liu et al., 2023) exchanges the temporal and variable dimension for MTS forecasting. The hidden dimension is 512. The learning rate is $1 \times 10^{-3}$.

**FourierGNN** (Yi et al., 2023) views all observed values as nodes in a graph, and perform graph learning in frequency domain to forecast MTS. The hidden dimension is 512. The learning rate is $1 \times 10^{-3}$.

**Mamba** (Gu & Dao, 2024) is a selective state-space model for sequence modeling. The hidden dimension is 128. The dimension for FCN is 16. The learning rate is $1 \times 10^{-4}$.

**TSMixer** (Chen et al., 2023) is an all-MLP model for MTS forecasting. The number of encoder layers is 2. The number of decoder layers is 1. The attention factor is 3. The hidden dimension is 256. The dimension for FCN is 512. The learning rate is $1 \times 10^{-4}$.

**PatchTST** (Nie et al., 2022) leverages patching in transformer for MTS forecasting. The patch lengths are 12, 90, 6, 300 and 10 for MIMIC-III, MIMIC-IV, PhysioNet'12, Human Activity, and USHCN, respectively. The number of encoder layers is 3. The attention factor is 3. The number of heads in attention is 16. The learning rate is $1 \times 10^{-4}$.

**Leddam** (Yu et al., 2024) uses learnable seasonal-trend decomposition for MTS forecasting. The hidden dimension is 512. The learning rate is $1 \times 10^{-3}$.

**BigST** (Han et al., 2024) uses a spatio-temporal graph neural network for traffic forecasting. The hidden dimension is 32. The dropout rate is 0.3. The learning rate is $1 \times 10^{-3}$.

**Reformer** (Kitaev et al., 2019) is an efficient implementation of the Transformer. The number of encoder layers is 2. The number of decoder layers is 1. The attention factor is 3. The learning rate is $1 \times 10^{-4}$.

**Informer** (Zhou et al., 2021) is another efficient implementation of the Transformer for MTS forecasting. The hidden dimension is 512. The learning rate is $1 \times 10^{-4}$.

**Crossformer** (Zhang & Yan, 2022) learns cross-dimensional dependencies for MTS forecasting. The segment lengths are 12, 360, 6, 300, and 3 for MIMIC-III, MIMIC-IV, PhysioNet'12, Human Activity, and USHCN respectively. The hidden dimension is 512. The learning rate is $1 \times 10^{-3}$.

A.4.2. METHODS FOR IMTS

**PrimeNet** (Chowdhury et al., 2023) is an IMTS pretraining model. Since the provided weights are specific to datasets with 41 variables, we retrain the model on all datasets. The patch lengths are 12, 180, 6, 300, and 10 for MIMIC-III,

MIMIC-IV, PhysioNet'12, Human Activity, and USHCN respectively. The number of heads in attention is 1. The learning rate is $1 \times 10^{-4}$.

**SeFT** (Horn et al., 2020) is a set-based method for IMTS classification. The number of layers is 2. The dropout rate is 0.1. The learning rate is $1 \times 10^{-3}$.

**mTAN** (Shukla & Marlin, 2020) converts IMTS to reference points for IMTS classification. The number of reference points are 32 on MIMIC-III and 8 on the rest datasets. The learning rate is $1 \times 10^{-3}$.

**NeuralFlows** (Biloš et al., 2021) is an efficient alternative to Neural ODE in IMTS analysis. The number of flow layers is 2. The latent dimension is 20. The hidden dimension for time is 8. The number of hidden layers is 3. The learning rate is $1 \times 10^{-3}$.

**CRU** (Schirmer et al., 2022) uses continuous recurrent units for IMTS analysis. The hidden dimension is 20. The learning rate is $1 \times 10^{-3}$.

**GNeuralFlow** (Mercatali et al., 2024) enhances NeuralFlows with graph neural networks for IMTS analysis. The flow model uses ResNet. The number of flow layers is 2. The latent dimension for input is 20. The latent dimension for time is 8. The number of hidden layers is 3. The learning rate is $1 \times 10^{-3}$.

**GRU-D** (Che et al., 2018) adapts GRUs for IMTS classification. The hidden dimension is 512. The learning rate is $1 \times 10^{-3}$.

**Raindrop** (Zhang et al., 2021) models time-varying variable dependencies for IMTS classification. The hidden dimension is 512. The learning rate is $1 \times 10^{-4}$.

**tPatchGNN** (Zhang et al., 2024) processes IMTS into patches and use graph neural networks for IMTS forecasting. The patch lengths are 12, 360, 6, 300, and 10 for MIMIC-III, MIMIC-IV, PhysioNet'12, Human Activity, and USHCN, respectively. The number of heads in attention is 1. The learning rate is $1 \times 10^{-3}$.

**Warpformer** (Zhang et al., 2023a) uses warping for multiscale modeling in IMTS classification. The hidden dimension is 64. The number of heads is 1. The dropout rate is 0. The number of layers is 3. The learning rate is $1 \times 10^{-3}$.

**GraFITi** (Yalavarthi et al., 2024) uses bipartite graphs for IMTS forecasting. The latent dimension is 128. The number of heads in attention is 4. The numeber of layers is 2. The learning rate is $1 \times 10^{-3}$.

