# OpenReview forum: "HyperIMTS: Hypergraph Neural Network for Irregular Multivariate Time Series Forecasting"
_ICML.cc/2025/Conference — ICML 2025 poster_

### Official Review · Reviewer_Fd4V · 2025-02-24

**Overall Recommendation:** 3

**Summary:**

The paper introduces HyperIMTS, a Hypergraph neural network for forecasting Irregular Multivariate Time Series (IMTS), which are challenged by irregular time intervals and unaligned observations. Existing models either use padded samples, disrupting sampling patterns, or set function and bipartite graphs, which struggle with unaligned dependencies. HyperIMTS addresses these issues by treating observed values as nodes in a hypergraph, with temporal and variable hyperedges enabling message passing. This irregularity-aware message passing captures both temporal and variable dependencies in a time-adaptive manner, improving forecasting accuracy. Experimental results show that HyperIMTS has good performance and low computational cost.

**Claims And Evidence:**

There are no problematic main claims.

**Essential References Not Discussed:**

No.

**Experimental Designs Or Analyses:**

The experiments in the paper are sufficient, with appropriate comparison methods and thorough analysis. The use of commonly employed real-world datasets enhances the credibility of the results.

**Methods And Evaluation Criteria:**

The model designed in the paper is both reasonable and effective for addressing the IMTS modeling problem. The dataset used is a commonly employed real-world dataset in IMTS modeling, which also provides practical guidance for solving real-world problems.

**Other Comments Or Suggestions:**

No.

**Other Strengths And Weaknesses:**

1. **The motivation of the paper is not persuasive enough**: The analysis of existing methods’ issues is not very convincing. For example, the authors claim that while padding increases the amount of data the model must process, this is not a critical issue, especially if the padded data is highly accurate, as it could provide more useful information for IMTS with high missing rates. Additionally, the authors emphasize the requirement for alignment in existing methods but then focus on the relationships between observations at the same time point $ t $ as an improvement, but this approach also requires aligned observation points between sequences.
2. **Some expressions in the paper are vague or incomplete**:

    a) The definition of padding-based methods is unclear. According to the authors, padding includes directly filling in real values,  RNN-based models that learn discrete hidden states and ODE-based models that learn continuous hidden states. However, there are significant differences between these approaches, and classifying them together is somewhat oversimplified.

    b) The classification of existing methods before the related work section is not comprehensive.  I recommend revising this description to avoid misleading researchers who are not familiar with this specific field.

**Questions For Authors:**

The method discussed in the paper is non-graph-based, but it simultaneously considers inter- and intra-series relationships. What is the main difference, aside from the update of correlations, compared to models that focus on these relationships?

**Relation To Broader Scientific Literature:**

This paper details efforts to advance IMTS forecasting across various scientific domains. The paper provides a new perspective on the forecasting problem of ISMTS to some extent and effectively improves forecasting performance.

**Theoretical Claims:**

There are no theoretical claims in this paper.

---

> ### Author Rebuttal · Authors · 2025-03-30
>
> Thanks for your careful reading and in-depth thinking. We address the concerns as follows:
> ## W1. Regarding highly accurate padding
> **A1.** We acknowledge that padding has pros and cons, which should be weighed up in IMTS forecasting.
> ### 1. **Error accumulation**
> Imputation errors can be amplified during subsequent predictions, as noted in previous works [3][4].
> We conduct additional experiments to verify it:
> - Pretrain each model for imputation.
> - Pad IMTS samples into fully observed via the imputation model.
> - Train the forecasting model on imputed samples.
>
> Better results are in **bold**:
> ||MIMIC III|PhysioNet'12
> |---|---|---
> |GraFITi|0.535(**33ms**)|**0.377(34ms)**
> |GraFITi(Impute)|**↑0.520**(245ms)|↓0.465(75ms)
> |Raindrop|**0.671(75ms)**|**0.438(52ms)**
> |Raindrop(Impute)|↓0.962(211ms)|↓0.820(85ms)
> |Crossformer|**0.641(184ms)**|0.408(**80ms**)
> |Crossformer(Impute)|↓0.962(292ms)|**↑0.265**(118ms)
>
> Over half of the results show **performance degradation** after imputation, while consuming **longer training time.**
>
> Although the experiment uses only a subset of models and datasets, we can still conclude that **existing models are not robust enough to provide highly accurate padding values**.
> ### 2. **Efficiency**
> The goal of our work is to improve forecasting accuracy **while maintaining high efficiency**.
> However, the above experiments demonstrate significant efficiency degradation.
> ### 3. **Informative missingness**
> As mentioned in Section 1 of Raindrop [2]:
> > the **absence of observations can be informative** on its own and thus imputing missing observations is not necessarily beneficial
>
> Absence actually reflects the **sampling pattern** of variables.
>
> E.g., in dataset PhysioNet'12, ALP and ALT, both liver function markers, can be observed to have nearly identical timestamps across all samples, indicating a *scheduled* sampling pattern.
> ## W2. Alignment in padding-based methods V.S. Our approach
> **A2.** We apologize for any confusion with respect to alignment. Revisions are made to clarify differences:
> ||Padding-based|Ours
> |---|---|---
> |*where* to align|data preprocessing & subsequent|only in time-aware similarity
> |*how* to align|padding|select shared timestamps only, or padding
>
> Alignment itself is not a problem, but *where* and *how* to align can **affect efficiency**.
> Our method aligns only in calculation of time-aware similarity by selecting shared timestamps, thereby eliminating the need to handle large padded data in other network modules.
>
> ## W3. Definition of padding-based methods
> **A3.** We apologize for not fully clarifying the definition and scope of padding.
> In this work, padding involves adding values, either zeros or predicted values (imputation), **in the sample space** before input time series are fed into the neural network, rather than within the neural network's latent space. E.g.:
> ||RNN-based|ODE-based
> |---|---|---
> |*latent* space|discrete|continuous
> |*sample* space (input)|zero-padded|zero-padded
>
> Further details on zero-padded input can be found in models' `forward` functions within the anonymized code repository provided at the end of abstract.
> ## W4. More comprehensive classification of existing methods
> **A4.** Thanks for your helpful advice!
> We revise them with the following improvements:
> 1. **Definition and scope of padding-based**: Clarify the definition scope is in sample space for input time series, instead of latent space.
> 2. **Classify methods from different perspectives**:
>     - **model architecture**: (1) RNN-based; (2) ODE-based; (3) GNN-based; (4) Set-based; (5) Transformer-based.
>     - **sample space padding methods**: (1) time-aligned padding; (2) patch-aligned padding; (3) non-padding.
> 3. **Compare with the standard in existing works**: Clarify the differences in classification standard with existing researches [5].
> ## Q1. Regarding model difference
> **A5.**
> - Our method is implemented as a **hypergraph attention network**, summarized in A3 within rebuttal to Reviewer VDUB above.
> - Main difference:
>     1. efficiency: light-weight architecture
>     2. performance: (1) aware of sample-specific alignment; (2) learn both relationships in parallel instead of sequentially, better capturing time-variable correlations.
>
> [1] Silva, Ikaro, et al; "Predicting In-Hospital Mortality of ICU Patients: The PhysioNet/Computing in Cardiology Challenge 2012"; Comput Cardiol (2010)
>
> [2] Zhang, Xiang, et al; "Graph-Guided Network for Irregularly Sampled Multivariate Time Series"; ICLR 2022
>
> [3] Wu, Sifan, et al; "Adversarial Sparse Transformer for Time Series Forecasting"; NeurIPS 2020
>
> [4] Ma, Qianli, et al; "Adversarial Joint-Learning Recurrent Neural Network for Incomplete Time Series Classification"; TPAMI 2020
>
> [5] Shukla, Satya Narayan, et al; "A Survey on Principles, Models and Methods for Learning from Irregularly Sampled Time Series"; arXiv:2012.00168
>
> ---
> **Thank you so much for helping us improve the paper! Please let us know if you have any further questions:-).**

---

### Official Review · Reviewer_Wd2p · 2025-02-27

**Overall Recommendation:** 3

**Summary:**

Irregular Multivariate Time Series (IMTS) are challenging due to irregular time intervals within variables and unaligned observations across variables, making it difficult to model temporal and variable dependencies. Existing IMTS models either use padded samples (which can introduce inefficiencies and distort sampling patterns) or represent data as bipartite graphs or sets (which struggle to capture dependencies among unaligned observations).

To address these limitations, the authors propose HyperIMTS, a Hypergraph Neural Network for IMTS forecasting. In this approach, observed values are treated as nodes in a hypergraph, with temporal and variable hyperedges facilitating message passing among all observations. This irregularity-aware message passing allows the model to capture variable dependencies in a time-adaptive manner, leading to improved forecasting accuracy. Experimental results show that HyperIMTS outperforms state-of-the-art models while maintaining low computational costs.

## update after rebuttal
Thank you for your response.

After reviewing the rebuttal, my original concerns remain largely unaddressed. First, I believe that set-based methods are still capable of capturing correlations, as the embeddings of independently processed observations are ultimately aggregated to produce the final output. Second, as acknowledged by the authors in the rebuttal, the proposed model relies on a shared feature medium (hyperedge), like a bipartite graph that depends on a shared temporal medium (time nodes).

That said, I recognize the novelty and potential impact of this work on the time-series literature, particularly in its approach to handling irregular time series. Therefore, I maintain my positive score.

**Claims And Evidence:**

The authors state that "However, sets (Set-based methods) typically do not account for correlations among observations." Can you provide more explanation why the set-based method fails to capture correlations among observations?

The authors state that "bipartite graphs are unable to propagate messages between variables without shared timestamps.". However, I think the proposed method has a similar limitation because hyperedge (feature information) is required to capture temporal dependencies.

**Essential References Not Discussed:**

No.

**Experimental Designs Or Analyses:**

I reviewed the experimental results in this paper. The inclusion of numerous baselines enhances the robustness of the findings, and the proposed method demonstrates strong performance.

**Methods And Evaluation Criteria:**

I think the proposed method makes sense for the suggested purpose.

**Other Comments Or Suggestions:**

N/A

**Other Strengths And Weaknesses:**

N/A

**Questions For Authors:**

N/A

**Relation To Broader Scientific Literature:**

I believe this method has a significant impact on the time-series literature, as it introduces a novel approach for handling irregular time series models and establishes a unified benchmark for irregular time series forecasting.

**Theoretical Claims:**

N/A

---

> ### Author Rebuttal · Authors · 2025-03-30
>
> Thanks for your review and encouraging feedback. We address the main concerns as follows:
> ## Q1. Why set-based method fails to capture correlations among observations?
> **A1:** We summarize the reasons from three aspects, using SeFT [1] as an example:
> ### 1. **Experimental aspect**
> - **Complexity**
>
>     As mentioned **in Section 3.3 of the SeFT paper**:
>
>     > "By contrast, our approach computes the embeddings of set elements **independently**, leading to lower runtime and memory complexity of $\mathcal{O}(n)$".
>
>     SeFT is compared with vanilla Transformers that have an $\mathcal{O}(n^2)$ complexity.
>
> - **Classification accuracy**
>
>     In the same section, the paper mentioned:
>     > "Furthermore, we observed that computing embeddings with information from other set elements (as the Transformer does) actually **decreases generalization performance** in several scenarios".
>
>     We note that this conclusion only applies to **classification** task, not **forecasting** task.
> ### 2. **Task aspect**
> SeFT was originally designed for classification rather than forecasting.
> The **task assumptions** used for time series classification and forecasting have minor differences:
> ||Classification|Forecasting|
> |---|---|---|
> |dependency *between* observations|Not necessary|Yes
> |*when* observations happened|Yes|Yes
>
> In classification, a few critical observations can be sufficient to determine the class label without considering their event order (e.g., a low heartbeat can signify potential health risks for patients); As for forecasting, future predictions depends on past observations, so their correlations are important.
>
> From experimental results, including dependencies between observations has been found to lower classification accuracy for SeFT, as mentioned previously.
> ### 3. **Modeling aspect**
> Set-based methods view observations as set elements, which are **invariant to their order**.
> The output of set functions doesn't change if observations are shuffled, which contrasts with models like RNNs or Transformers.
> ## Q2. Proposed method has similar limitation as bipartite graphs
> **A2:** We provide a detailed explanation of how our model learns variable dependencies without relying on shared timestamps.
> ### 1. **Preliminary**
> In our model:
> - **temporal dependencies**: learned via message propagations among observations *within the same variable*
> - **variable dependencies**: learned via message propagations among observations *between different variables*.
> ### 2. **Our model**
> Variable dependency learning is conducted through a **three-step** process.
>
> In Figure 2 of our work, variables $V_1$ and $V_3$ do not have shared timestamps.
> When propagating messages from observation $(t_1, V_1)$ (i.e., the blue "1" node) to $(t_2, V_3)$ (i.e., the orange "2" node), the process is:
> 1. **node to hyperedge**: node $(t_1, V_1)$ $\rightarrow$ variable hyperedge $V_1$
> 2. **hyperedge to hyperedge**: variable hyperedge $V_1$ $\rightarrow$ variable hyperedge $V_3$
> 3. **hyperedge to node**: variable hyperedge $V_3$ $\rightarrow$ node $(t_2, V_3)$
> ### 3. **Bipartite graph**
> In contrast, the bipartite graph method propagates messages in a **two-step** manner.
> As depicted in Figure 1 (d), for variable dependencies, messages can only be propagated between $V_1$ and $V_2$ (*with* shared timestamps $t_1, t_2, t_3$), but not $V_1$ and $V_3$ (*without* shared timestamps).
> We describe the process of message propagation between $V_1$ and $V_2$ (*with* shared timestamps) for additional clarity:
> 1. **variable node to time node**: variable node $V_1$ $\rightarrow$ time node $t_1$ (or $t_2, t_3$)
> 1. **time node to variable node**: time node $t_1$ (or $t_2, t_3$) $\rightarrow$ variable node $V_2$
>
> To summarize, bipartite graph relies on shared time nodes to propagate messages among variables, while our method HyperIMTS uses variable hyperedge interactions instead, which bypasses the requirement of shared timestamps.
>
> [1] Horn, Max, et al; "Set Functions for Time Series"; ICML 2020
>
> ---
> **Lastly, thanks again for your careful review and appreciation! Feel free to let us know if you have any further questions or concerns :-).**

---

### Official Review · Reviewer_VDUB · 2025-03-20

**Overall Recommendation:** 3

**Summary:**

The paper introduces a hypergraph neural network designed for forecasting irregular multivariate time series, which are characterized by irregular time intervals within variables and unaligned observations across variables.

**Claims And Evidence:**

While the claims are generally supported, more comprehensive efficiency reporting and statistical analysis of performance differences would strengthen the evidence.

**Essential References Not Discussed:**

N/A

**Experimental Designs Or Analyses:**

The experimental design is robust and sound.

**Methods And Evaluation Criteria:**

The proposed methods align well with the IMTS forecasting problem.

**Other Comments Or Suggestions:**

None.

**Other Strengths And Weaknesses:**

Strengths:
(1) Novel application of hypergraphs to IMTS forecasting, creatively combining existing ideas.
(2) Avoids padding, reducing computational overhead, as shown in Figure 4.
(3) Extensive comparison with 27 baselines across five datasets is a significant effort.

Weaknesses:
(1) Compared to the existing works, the hypergraph structure modeling may be challenging to implement and optimize practically.
(2) As noted in Section 6, it cannot handle text or images, restricting applicability in some domains.
(3) Resource-intensive compared to state-space models like Mamba.

**Questions For Authors:**

see weakness.

**Relation To Broader Scientific Literature:**

HyperIMTS relates to prior work such as IMTS modeling, GNNs for time series, and HGNNs.

**Theoretical Claims:**

The paper does not present formal theorems or proofs. Its theoretical foundation lies in the hypergraph structure and message-passing mechanisms, drawing from graph theory and neural network principles.

---

> ### Author Rebuttal · Authors · 2025-03-30
>
> Thanks for your review and valuable advice. We address the concerns as follows:
> ## W1. Efficiency reporting and performance analysis
> **A1:** Additional efficiency analyses can be found in **Appendix A.3**, and performance analyses on varying lookback lengths and forecast horizons are detailed in **Appendix A.2**, which can be summarized as:
> ### 1. **Insensitive to time length**
> Non-padding methods can handle longer time lengths with **smaller efficiency degradation**, compared to padding-based methods.
> - In Figure 6, non-padding methods **maintain high efficiency** for longer time lengths.
> - In Figure 7, although non-padding methods may not be the fastest on datasets with short lengths (47 timestamps for PhysioNet'12, 131 timestamps for Human Activity), they are **among the fastest for long input lengths** (971 timestamps for MIMIC-IV, 337 timestamps for USHCN).
> ### 2. **Best-performing on various lengths**
> - In Table 5, our method remains the **best-performing** at longer forecast horizons compared to the strongest baselines, where 5 out of 10 models are within 1 year [1-5].
> - In Figure 5, our method has the **lowest MSE** in 13 out of 15 lookback length settings.
> ## W2. Codes and hyperparameter settings
> **A2:** We apologize for any clarity issues in presenting our code and hyperparameter settings.
> ### 1. **Code snippets**
> An anonymized repository link is available **at the end of the abstract, line 40**. An overview of the codes can be found in answer **A3** below.
> ### 2. **Hyperparameter settings**
> They are discussed in **Appendix A.4** and provided in the 'scripts' directory within the anonymized code repository.
> We follow the hyperparameter settings **from original papers or codes** for baselines, except for batch size (32), max epoch (300), and early stopping patience (10).
> The differences in critical hyperparameter settings for top-performing models on MIMIC-III are shown here:
> ||Ours|GraFITi|Warpformer|Crossformer|
> |---|---|---|---|---|
> |seg length|-|-|-|12|
> |hidden dimension|128|128|64|128
> |attention heads|4|4|1|8|
> |drop out|-|-|0|0.05
> ## W3. Challenging to implement and optimize practically
> **A3:** Code implementation of our model is provided **in `models/HyperIMTS.py` file within the anonymized code repository**.
> A summary of the code implementation is provided here:
> ### 1. **IMTS to hypergraph**
> `HypergraphEncoder` class, which outputs:
> - observation nodes: Eq.3, linear layers + ReLU.
> - temporal hyperedges: Eq.4, sinusoidal encoding.
> - variable hyperedges: Eq.5, learnable parameters.
> - incidence matrices:
>     Eq.2, indicate for every hyperedge, which observation nodes are connected to it.
> ### 2. **Hypergraph learning**
> `HypergraphLearner` class, which is a **hypergraph attention network** similar to the backbone used in previous irregular time series models such as GraFITi [3] and Raindrop [6]:
> - node-to-hyperedge: Figure 3 (a), `MultiHeadAttentionBlock` class.
> - hyperedge-to-hyperedge: Figure 3 (b), `IrregularityAwareAttention` class.
> - hyperedge-to-node: Figure 3 (c), mainly includes self attention and linear layers with activations.
> ### 3. **Hypergraph to IMTS**
> Eq.15, A linear layer that maps from the hidden space to the sample space.
> ### 4. **Optimization**
> The core module `HypergraphLearner` consists of **only 2 layers with residual connections** around attention operations, achieving its best performance  **within 50 epochs** across all datasets in our experiments.
> ## W4. Cannot handle text or images; Resource-intensive compared to SSMs
> These possible improvements are not included for the following reasons:
> ### 1. **Unfair comparisons** (text/images)
> Most SOTA time series models only accept time series as input.
> If our model takes more input data than others, it would be unfair for these baselines.
> ### 2. **Untested with HGNNs** (SSMs)
> Graph attention networks have been shown to be reliable in existing works [3][6].
> However, SSMs have not yet been tested with HGNNs for time series yet, which might lead to unexpected issues like performance degradation.
>
> [1] Mercatali, Giangiacomo, et al; "Graph Neural Flows for Unveiling Systemic Interactions Among Irregularly Sampled Time Series"; NeurIPS 2024
>
> [2] Shang, Zongjiang, et al; "Ada-MSHyper: Adaptive Multi-Scale Hypergraph Transformer for Time Series Forecasting"; NeurIPS 2024
>
> [3] Yalavarthi, Vijaya Krishna, et al; "GraFITi: Graphs for Forecasting Irregularly Sampled Time Series"; AAAI 2024
>
> [4] Zhang, Weijia, et al; "Irregular Multivariate Time Series Forecasting: A Transformable Patching Graph Neural Networks Approach"; ICML 2024
>
> [5] Liu, Yong, et al; "iTransformer: Inverted Transformers Are Effective for Time Series Forecasting"; ICLR 2024
>
> [6] Zhang, Xiang, et al; "Graph-Guided Network for Irregularly Sampled Multivariate Time Series"; ICLR 2022
>
> ---
> **We hope the above response can help solve your questions. Thanks again for your thorough review and looking forward to your reply!**

---

### Decision · Program_Chairs · 2025-05-01

**Decision:**

Accept (poster)

**Comment:**

The idea of using hypergraphs to directly model irregular multivariate time series is both refreshing and practically useful. Despite some concerns about limited generalizability to other modalities and the complexity of implementation, the reviewers consistently agree the method is well-motivated and shows strong empirical results across five diverse datasets. The unified design, thoughtful ablations, and comprehensive comparisons make a compelling case. I believe this work makes a meaningful contribution and is ready for acceptance.